# Upcycling and Recycling Potential of Selected Lignocellulosic Waste Biomass

**DOI:** 10.3390/ma14247772

**Published:** 2021-12-16

**Authors:** Anita Wronka, Eduardo Robles, Grzegorz Kowaluk

**Affiliations:** 1Institute of Wood Sciences and Furniture, Warsaw University of Life Sciences—SGGW, Nowoursynowska St. 159, 02-776 Warsaw, Poland; anita_wronka@sggw.edu.pl; 2University of Pau and the Adour Region, E2S UPPA, CNRS, Institute of Analytical and Physicochemical Sciences for the Environment and Materials (IPREM-UMR 5254), 40004 Mont de Marsan, France; eduardo.robles@univ-pau.fr

**Keywords:** biopolymer, wood, upcycling, composite, recycling, mechanical properties, physical properties, carbon storage, raspberry, black chokeberry, bio waste

## Abstract

This research aimed to confirm the ability to reduce carbon dioxide emissions by novel composite production using plantation waste on the example of lignocellulosic particles of black chokeberry (*Aronia melanocarpa* (Michx.) Elliott) and raspberry (*Rubus idaeus* L.). Furthermore, to characterize the particles produced by re-milled particleboards made of the above-mentioned alternative raw materials in the light of further recycling. As part of the research, particleboards from wooden black chokeberry and raspberry were produced in laboratory conditions, and select mechanical and physical properties were examined. In addition, the characterization of raw materials (particles) on the different processing stages was determined, and the fraction share and shape of particles after re-milling of the produced panels was provided. The tests confirmed the possibility of producing particleboards from the raw materials used; however, in the case of boards with raspberry lignocellulose particles, their share cannot exceed 50% so as to comply with the European standards regarding bending strength criterion. In addition, the further utilization of chips made of re-milled panels can be limited due to the significantly different shape and fraction share of achieved particles.

## 1. Introduction

Every year, society’s awareness of caring for the Earth is growing. The growing amount of waste is a problem, with its storage and greater carbon dioxide emissions. In the case of fruit bushes grown in Poland, which are pruned each year, their branches are often left in the field or are burned. It can be used as a biofuel to avoid wasting energy, but it is not yet a common practice in Poland. Another way to use orchard waste is to produce three-layer particleboards for the furniture industry. Even though the tree species used for wood products are renewable, it should not be limited only to it because renewable does not mean that it is infinite. Because of this, it is necessary to explore using other lignocellulosic materials that will fully or partially replace the wood raw material. This attempt to move into the broad utilization of renewable biopolymers was also suggested by Bari and collaborators [1]. Some attempts have already been made that have been proven to be more or less effective, for this purpose, materials such as pepper stalks [2], sugarcane [3,4], almond shell [5], apple and plum branches [6], bamboo chips [7], straw [8], wheat straw and corn pith [9], kiwi prunings [10], coffee husk [11], flax shiv [12], acai (*Euterpe oleracea* Mart.) fruit [13], oil palm empty fruit bunch [14], and kenaf [15] were used. The use of wooden lignocellulosic parts of fruit plant waste allows for the binding of carbon dioxide in the form of particleboards, without emitting it into the atmosphere. In this field, good examples are raspberry (*Rubus idaeus* L.) and chokeberry (*Aronia melanocarpa* (Michx) Elliott) plantation waste. These represent substantial waste in Polish fields, as Polish production accounts for 60–70% of the world’s production potential. The cultivation area of chokeberry is about 40 km^2^ per year and the annual harvest of fruits is from 40 to 60 thousand tonnes. The main recipients of chokeberry fruits are China, Japan, and South Korea [16]. Whereas the area of raspberry cultivation in Poland is over 290 km^2^, placing it at fifth place in the world’s raspberry producers and third in Europe (after Russia and Serbia) [17].

The fruits of these shrubs are cultivated for their taste and health benefits. Raspberry fruits are rich in anthocyanins and have anti-inflammatory and anticancer properties, so it is often recommended to drink raspberry juice during colds [18]. Medicinal values also characterize black chokeberry fruits; just like raspberries, they have an antioxidant effect, and their consumption is recommended to prevent chronic diseases [19]. This added value for the fruits allows for assuming that the potential availability of lignocellulosic resources of those above-mentioned alternative raw materials will grow shortly. Therefore, it seems worth researching the development of long-term storage regarding the carbon fixed in these raw materials, such as producing particleboards and attempting to upcycle these wooden wastes and recycle the produced composites.

This investigation aimed to determine the ability to utilize raspberry and chokeberry lignocellulosic particles to produce particleboards for furniture purposes and to characterize the wooden particles produced by the re-milling particleboards mentioned regarding further recycling. As a result, the following hypothesis has been investigated: the lignocellulosic particles of raspberry and black chokeberry are valuable raw materials to produce the particleboards and obtain particles from re-milled panels, which can potentially be re-used in particleboard production.

## 2. Materials and Methods

### 2.1. Materials

Raspberry (*Rubus idaeus* L.) (Figure 2) and black chokeberry (*Aronia melanocarpa* (Michx.) Elliott) (Figure 3) wooden stalks were used for the current work. Two year old raspberry stalks, as well as four year old chokeberry rods, were collected from Polish fields. The raw materials were dried in a chamber drier under 70 °C to air-dry the moisture content (about 10–12%), and the bark content (*w/w*) was measured by manual debarking about 2 kg of each tested material. The wooden branches of the chokeberry and raspberries were shredded on saw blade in separate batches (50 mm long chips) and then grounded into a fine fraction using a laboratory three-knife drum mill (laboratory prototype) with an outlet equipped with 6 × 12 mm^2^ mesh to form particles. The bulk density of the particles was calculated as the weight of a selected fraction, divided by the measuring cylinder’s capacity (in volume). The measurement was repeated five times for every fraction. The produced particles were sorted on mesh of size 0.5 and 1 mm (face layers), and 8 mm and 2 mm (core layer) to exclude the oversized and undersized particles. The pictures of the cross-cuts of the investigated raw materials were taken with a NIKON SMZ 1500 (Kabushiki-gaisha Nikon, Minato, Tokyo, Japan) optical microscope.

### 2.2. Elaboration of Composites

Three-layer composites were produced as particleboards (PB) with different black chokeberry and raspberry contents. The lignocellulosic particles were dried to a moisture content of 5%. As a result, particleboards with a nominal density of 600 kg m^−3^, 32% (*w/w*) of face layer content, and a total thickness of 16 mm were produced. The following content (*w/w*) of alternative raw materials was applied: 0% (reference panels, 100% of industrial (coniferous) particles), 10%, 25%, 50%, and 100%. The industrial urea-formaldehyde resin Silekol S-123 (Silekol Sp. z o.o., Kędzierzyn—Koźle, Poland) was used to resinate the particles, where the resination of particles for the face layer was 12% and the core layer was 10%. No hydrophobic agent (like paraffin emulsion) was added. The curing was done for 82 s inside an oven at 100 °C. Panels were pressed on a hydraulic press (ZUP-NYSA PH-1P125) at a maximum pressure of 2.5 MPa, with a temperature of 200 °C, and a time factor of 20 s mm^−1^. The produced boards were conditioned before the tests in a climatic chamber (producer: Research and Development Centre for Wood-Based Panels Sp. z o. o. in Czarna Woda, Poland) at 20 °C and 65% air humidity, until a constant mass was obtained. The main steps of the material flow and samples preparation are presented in Figure 1.

### 2.3. Characterization of the Elaborated Panels

All of the elaborated PB were conditioned at 20 °C, and the test specimens were cut on a saw blade, as required by European standards EN-326-2 [20] and EN-326-1 [21]. The modulus of rupture (MOR) and elasticity (MOE) were determined according to EN 310 [22], and the internal bond (IB) was determined according to EN 319 [23]. All the mechanical properties were examined with an INSTRON 3369 (Instron, Norwood, MA, USA) laboratory-testing machine, and, whenever applicable, the results were referred to standards [24]. Board density was determined according to EN 323 [25], thickness swelling (TS) to EN 317 [26], and surface water absorption was done following EN 381-1 [27]. The density profiles of the tested PB (three types: reference, 100% of raspberry, and 100% of chokeberry) were measured on a GreCon DAX 5000 device (Fagus-GreCon Greten GmbH and Co. KG, Alfeld/Hannover, Germany). 

### 2.4. Raw Material Recycling and Characterization

The composites were re-milled on a laboratory knife mill (laboratory prototype delivered by Research and Development Centre for Wood-Based Panels Sp. z o. o. in Czarna Woda, Poland) equipped with three knives, two contra-knives, and a 6 × 12 mm^2^ mesh. The fraction of chips taken from the re-milled particleboards was tested with an IMAL (Imal s.r.l., San Damaso (MO), Italy) vibrating laboratory sorter with seven sieves. The selected sieve sizes were 8, 4, 2, 1, 0.5, 0.25, and <0.25 mm. The amount of tested material for each fraction was about 100 g, and the set time of continuous vibrating was 5 min. As many as five repetitions were done for every tested material.

### 2.5. Statistical Analysis

Analysis of variance (ANOVA) and t-tests calculations were used to test (α = 0.05) for significant differences between factors and levels, where appropriate, using IBM SPSS statistic base (IBM, SPSS 20, Armonk, NY, USA). A comparison of the means was performed when the ANOVA indicated a significant difference by employing the Duncan test. The statistically significant differences for the achieved results are given in the Results and Discussion paragraph whenever the data were evaluated.

## 3. Results and Discussion

### 3.1. Materials Characterization

The bark content (*w/w*) was 7.4% for raspberry and 18.0% for chokeberry. According to [28], the average bark content of pine (*Pinus sylvestris* L.), which is the main raw material for particleboard production in Poland, is about 6.7% (*w/w*). Significant differences in bark content of the investigated materials were found. Such a high content of bark in the case of chokeberry could influence the mechanical properties of the produced PB [29]. It was found by Kowaluk et al. [6] that the bark density of orchard trees can be remarkably lower than the density of the wood. As confirmed in the case of single-layer particleboards produced from *Quercus cerris* bark [30], these panels had remarkably low mechanical properties when compared to the commercial particleboards. What was also confirmed by the mentioned researchers, is that the panels produced from *Quercus cerris* bark had low TS. The bark particles, being highly brittle, could also raise the fine particles/dust production when milled, which could negatively influence the mechanical properties of the panels. On the other hand, a fine bark particle can be upcycled and utilized, as was proven by Mirski and collaborators [31].

Concerning the anatomy of the investigated raw materials, raspberry (Figure 2) has a large amount (volumetric) of foamy parenchyma pith. However, this part of the material can be easily disintegrated mechanically, and it is not easy to separate the particles produced from the remaining particles. Furthermore, as a brittle and soft tissue, it produces a large amount of fine particles, characterized by a large specific surface. This feature is not desirable in PB production, since this fraction requires a high amount of resin to be added. If the resination is not tuned regarding these fine fractions, the mechanical parameter of produced PB drops down.

The cross-cut of chokeberry (Figure 3) can be referred to as broadleaf plants. The year rings (Figure 3a) are clearly visible, and wood rays are going horizontally between bark and pith (Figure 3b; bark on the left, pith on the right). It is worth pointing out that the pith is also in foam form, which was found for raspberry stalks, but here the amount of foam pith was significantly lower than for raspberry.

In Figure 4, the results of the measurement of the bulk density of particles used to produce the tested composites and those produced by re-milling of the tested composites are presented. In the case of the face layers’ intended particles, the highest bulk density was found for chokeberry particles (164 kg m^−3^). A 2.4% lower bulk density (when referred to highest value) was found for industrial face layer particles (160 kg m^−3^). The lowest bulk density value among the tested particles was registered for raspberry particles (83 kg m^−3^), which means an almost 50% lower density for chokeberry. When analyzing the core layer purpose particles, the results were as follows (descending order): industrial (157 kg m^−3^), chokeberry (121 kg m^−3^), and raspberry (89 kg m^−3^). The results of the measurement of the bulk density of re-milled particles show that the bulk density of these particles was higher than for the primary particles, and, what should be pointed out, is that the differences between the tested materials were less than 2% when considering the lowest value. The achieved average bulk density values were statistically significantly different when compared within the same group (face, core, and re-milled).

The achieved density values of raspberry and chokeberry particles were low compared to other alternative lignocellulosic raw materials [6]. This is promising information, as, in the case of compressed lignocellulosic composites, a low bulk density leads to better densification, creating more spots where separate particles are connected. Thus, the produced composite structure is more even, less porous, and has higher mechanical properties. This can also lead to lower water absorption. However, it was confirmed by Papadopoulos et al. [7] that a lower bulk density can reduce the mat permeability due to densification during hot pressing, and the heat transfer through such a mat can be significantly slower.

### 3.2. Modulus of Rupture and Modulus of Elasticity

As shown in Figure 5, the modulus of rupture values decreased when the content of alternative raw materials increased. The MOR decrease was higher for composites produced of raspberry particles (from over 15 N mm^−2^ when 0% of raspberry particles to less than 10 N mm^−2^ for 100% raspberry composite). In the same conditions, MOR decreased for chokeberry composites that had reached over 12.1 N mm^−2^. When compared within the same raw materials, the only statistically significant differences between average MOR values were found for the highest and lowest content of raw materials. When referring to the EN standard [24], it was found that in the case of raspberry, the content of alternative raw material should not exceed 50% for meeting the standard requirements.

Similar tendencies for a reduction of MOE when the alternative raw material contend grows are presented in Figure 6. The reduction of MOE from the value of reference composite, 2805 N mm^−2^, was to 1800 N mm^−2^ for the fully chokeberry composite and to 1617 N mm^−2^ for the 100% raspberry composite. It is worth adding that when referring to the EN standard [24], in the case of raspberry, the content of alternative raw material should not exceed about 80% for meeting the standard requirements. Furthermore, statistically significant differences for the average MOR values for chokeberry were found for all composite types excluding 25 and 50% alternative raw material particles share, when, in the case of raspberry, there were no statistically significant differences between the composites of 10 and 25%, as well as between 50 and 100%.

Raspberry panels were expected to have the best mechanical properties because of their lower bulk density; however, they presented low mechanical features for MOR and MOE. The reason for this can be the content of the core, as can be appreciated in Figure 2. This part of the raw material can influence the production of particles with a low bulk density, but these particles do not allow for carrying a high mechanical load when the samples are bent, the face layers are strained/compressed, and core layers are under shear stress. Moreover, the geometry of the particles used to produce the composites can play a role here. As Wronka and Kowaluk demonstrated [32], the raspberry particles are shorter and have blunt (wide) ends when compared to industrially used softwood particles. Furthermore, because of the structure of the raspberry stalk, where the region of higher mechanical properties is on the external zone of the rod, the particles produced from this raw material are of a lower length to thickness ratio (slenderness), which is not desirable for particle composites. It has been confirmed [33] that the best mechanical properties for composites are achievable with a high length to thickness ratio.

### 3.3. Internal Bond

The positive effect of a low bulk density of raspberry particles has been found when analyzing the IB values of the tested composites. As shown in Figure 7, the IB was significantly raised when the content raspberry particles rose. The reference composite IB value was 0.72 N mm^−2^, and, for 100% raspberry composite, the IB was 1.04 N mm^−2^, while for 100% chokeberry composite, it was 0.53 N mm^−2^. It should be pointed out that when comparing the achieved results of IB, all of the tested composites met the requirements of a specified European standard [24]. Furthermore, the statistical analyses within the alternative raw materials mentioned show no statistically significant differences between IB average values of 10% and 25%, 25%, and 50% for raspberry, as well as between 10% and 50%, and between 25% and 100% for chokeberry.

### 3.4. Thickness Swelling and Water Absorption

The results of the measurement of thickness swelling of the tested composites after 2 h and 24 h of soaking in water are presented in Figure 8. As can be seen, in the case of raspberry composites, the swelling in thickness significantly grew with the alternative raw material content increase. After 2 h of soaking, the lowest TS for the reference composite (0% of raspberry particles) was 18%, while for the 100% raspberry panel, the TS was 33%, which is an increase of more than 83%. After 24 h of soaking, the TS of the reference composite was below 20%, and for the 100% raspberry composite, the TS was over 36% (89% growth). In the case of chokeberry, the increase in alternative raw material particles content caused a decrease in thickness swelling. After 24 h of soaking of the chokeberry composites, the TS was 16%, which was an almost 16% reduction of TS. The only statistically significant differences for the average values of TS after 2 h of raspberry composites were found between the 0%, 50%, and 100% panels and the same composites after 24 h of soaking. Regarding chokeberry, statistically significant differences after 2 h of soaking were found for composites of 0% and 100%, and the same after 24 h of soaking. It should be highlighted that when referring to the achieved results of TS, none of the tested composites met the requirements of the European standard [24].

Such a significant rise of TS of composites made of raspberry particles can be explained by the low bulk density of raspberry particles, as presented in Figure 4. Although a low bulk density helps with better densification of the pressed mat, this highly compressed mat can be easily decompressed by water penetrating the composite in light of the swelling in thickness. Thus, the material, which was more densified during hot pressing (composite preparation), has a potential of higher TS. On the other hand, the opposite situation was found in the case of chokeberry composites, where the mat densification was lower due to the higher bulk density of the chokeberry particles.

The WA values of the tested composites of different contents of alternative raw materials are presented in Figure 9. The high water absorption values after 24 h of soaking for the raspberry samples, from over 77% for the reference composite to over 108% for 100% raspberry composite, can be explained by the presence of low-density core particles, which can react with water like a sponge. The higher increase of WA for the samples with a higher content of raspberry particles after 24 h compared to WA after 2 h of soaking means that the structure of the samples is less penetrative (tighter) against water, and more time is needed to reach the deeper zones of the samples. This can be explained by the higher densification of the mat built by low bulk density particles. When evaluating the WA of chokeberry composites, it can be found that with the rising content of chokeberry particles, the WA slightly rose after 2 h of soaking, whereas, after 24 h of soaking, the WA decreased with the increase in chokeberry particles content. This means that chokeberry particles cause lower water absorption. A specific type of composite here can be the 100% chokeberry panel, where the maximum WA was reached after 2 h of soaking and did not raise even after 24 h of total soaking. One of the reasons could be the high bulk density of chokeberry particles, which lead to lower compression of particles during pressing, and leave more unfilled (empty) zones in the composite structure. These zones can be filled with water in a short time. Another reason is that the deciduous wood has a five times higher potential to transfer the water due to the larger dimensions of the vessels [28]. Statistically significant differences of average WA for raspberry composites after 2 h of soaking were between 0% and 100%, and between 0%, 10%, and 25% against 50% and 100% composites after 24 h. For chokeberry, these differences were found between samples of 0%, 10%, and 25% against 100% after 2 h, and 0%, 10%, and 25% against 50% and 100% composites after 24 h.

### 3.5. Density and Density Profiles

The results of the density profile measurement of the tested composites are presented in Figure 10. Since the tested composites are symmetrical, half of the density profile is shown to improve the readability of the plots. As can be seen, the highest values of density in the face zone, over 950 kg m^−3^, located about 0.7 mm in deep from the surface, were found for the industrial particles composite. On the other hand, the highest density zone of the raspberry composite, about 815 kg m^−3^, was found about 1.8 mm under the composite surface. A similar zone for the chokeberry composite, but with a lower density, about 780 kg m^−3^, was found at 0.4 mm under the surface. In the case of the lowest density in the core layers (middle of the thickness), the lowest value, about 550 kg m^−3^, was registered for the raspberry composite, when the remaining composites had a similar core layer density, which was about 590 kg m^−3^. It should be mentioned here that all of the tested samples were of the same average density of about 600 kg m^−3^.

A high density of face layers, which was found for the reference composites (industrial particles), can significantly influence the bending properties of composites as these face layers are generally responsible for tension and compression stresses when the material is bent. This remark can be confirmed on figures presenting MOR (Figure 5) and MOE (Figure 6) values. It was also confirmed for fibrous composites of different density profiles [34]. However, as the differences between the density values of core layers of the tested panels are low, it can be hard to refer to the remaining features of the tested composites.

### 3.6. Recycled Material Characterization

The pictures of different particles produced by re-milling the tested composites and industrial (not re-milled) particles are shown in Figure 11. It can be found that in the case of industrial particles, for all fractions excluding dust (< 0.25 mm), the particles had a high length to width ratio, which can be estimated on the level of 4:1 for 8 mm fraction and even higher, about 20:1 for 2 mm fraction. On the other hand, the pictures of re-milled particles of fractions 8 and 4 mm show that the particles were not elongated anymore, and these were more rounded or square, with a length-to-width ratio of about 1:1. For a fraction of 2 mm, a significant difference was found for chokeberry particles, which are more similar to industrial particles. In addition, the smaller chokeberry particles (1 mm and below) are closer to industrial (not re-milled) particles. When re-milled, industrial and raspberry particles are short and of higher width. These remarks can be valuable in the light of further use of re-milled particles, since, as confirmed, the shape of the particles can significantly influence the properties of the produced particle composites [35,36].

The mass fraction share of particles produced by the re-milling of investigated composites and the fraction share of primary industrial particles is shown in Figure 12. As can be seen, in the case of raw industrial particles, the largest share is for particles of size 1 and 2 mm (over 74%), and 17% of size 4 mm. The remaining fractions are less than 9%. When analyzing the fraction share of re-milled particles, it can be stated that the fraction share of industrial and chokeberry particles is similar. The difference is between the distribution of fractions smaller and larger than 1 mm: for industrial re-milled particles, a more significant amount of fractions smaller than 1 mm were found, and a smaller amount of fractions were larger than 1 mm. The opposite distribution was found for chokeberry particles. Significant differences in fraction share were found for raspberry re-milled particles. These particles had many fractions of bigger dimensions, where the content of fractions of 4 mm + 8 mm was about 47%. This type of material also provides a large amount of smaller fractions: the sum of fractions 0.5 mm, 0.25 mm, and < 0.25 mm was about 30%, whereas, in the case of remaining re-milled materials, it was about 23% for industrial re-milled, 18% for re-milled chokeberry, and 8% for primary industrial particles. It should be pointed that a high amount of small fractions is not profitable when considering the achieved particles to be used as a raw material to produce similar particle composites. Since the specific surface of the particles grows with the particle size decrease, and, thus, a larger amount of binder is needed to cover the particle surface adequately, such small fractions should be separated and subjected to alternative processing/utilization.

## 4. Conclusions

According to the conducted research and the analysis of the achieved results, the following conclusions and observations can be drawn:It has been confirmed that lignocellulosic particles of black chokeberry (*Aronia melanocarpa* (Michx.) Elliott) and raspberry (*Rubus idaeus* L.), being an orchard waste, can be successfully upcycled and used to produce lignocellulosic composites, thus having a positive contribution to carbon storage.The bulk density of chokeberry particles on the outer layers is slightly higher than that of industrial particles. The inverse relationship occurs in the case of particles on the core layers. The particles in both layers are characterized by a lower density than the reference (industrial) particles for raspberry.With an increase in the proportion of black chokeberry or raspberry particles in the particleboard, the bending strength and modulus of elasticity decreases.A significant influence of the content of black chokeberry and raspberry particles was found on the perpendicular tensile strength (IB) of the tested composites: a significant increase with raspberry particles increasing and decrease with chokeberry particles increasing.The thickness swelling of raspberry-containing composites increases after 2 h and 24 h of soaking in water. In the same conditions, the increase of chokeberry particle contents causes a lower thickness swelling.The water absorption test showed increasing dynamics of water absorption for boards with a higher proportion of chokeberry and raspberry particles, but in the long run, boards made of chokeberry particles absorb less water than the reference and raspberry composites.The highest density of face layers has been found for reference composites made of industrial particles, which influence the bending features of the tested composites.Further use of particles produced from re-milled composites can be limited due to the shape of the re-milled particles, which, in the case of industrial and raspberry particles, is significantly different from unprocessed particles.

## Figures and Tables

**Figure 1 materials-14-07772-f001:**
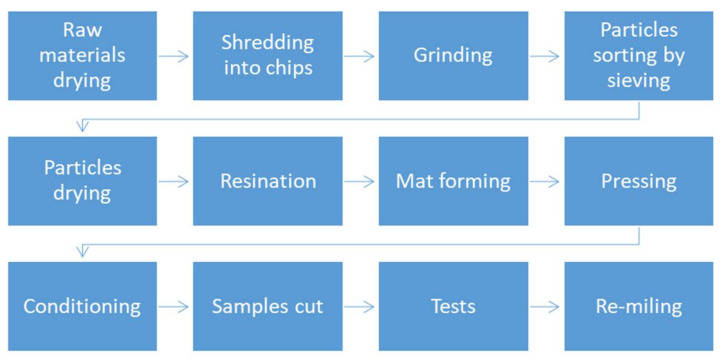
The process of material flow and samples preparation chart.

**Figure 2 materials-14-07772-f002:**
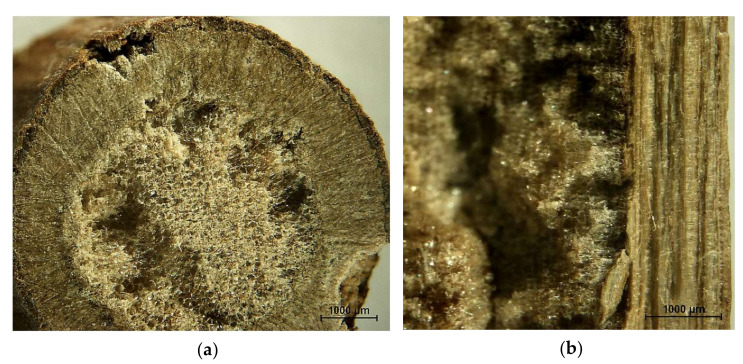
Cross-cut: (**a**) across and (**b**) along the fibers of a raspberry stalk.

**Figure 3 materials-14-07772-f003:**
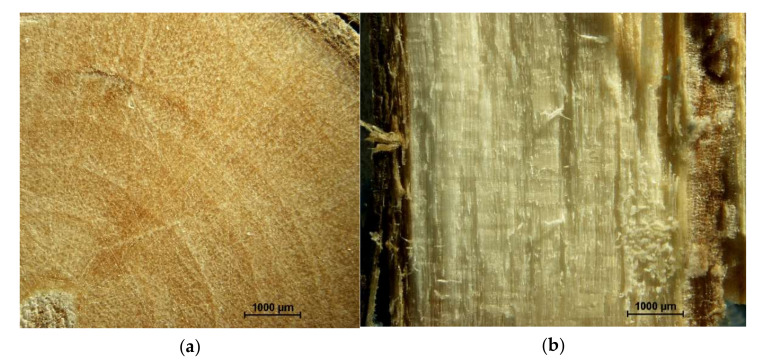
Cross-cut: (**a**) across and (**b**) along the fiber of the chokeberry stalk.

**Figure 4 materials-14-07772-f004:**
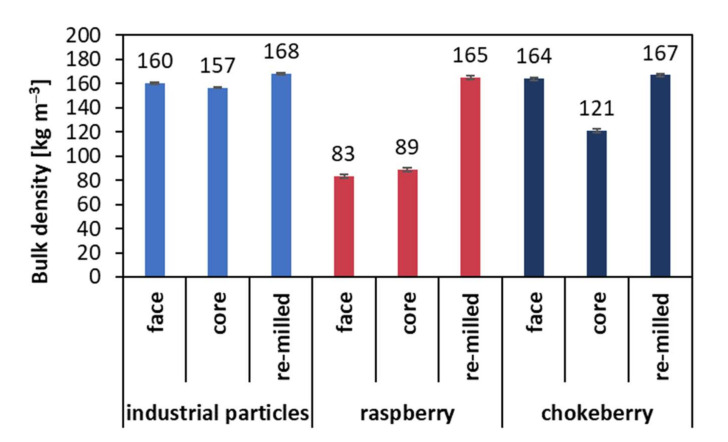
Bulk density of particles used to produce composites, considering the particles produced by the re-milling of composites.

**Figure 5 materials-14-07772-f005:**
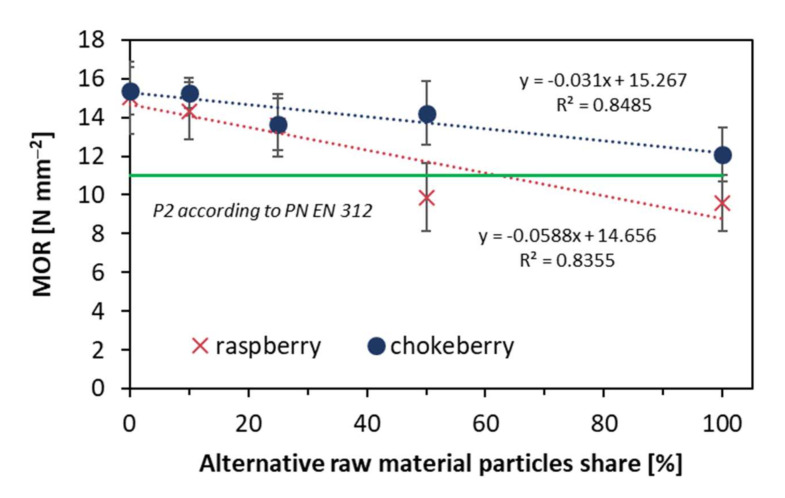
Modulus of rupture of the tested composites.

**Figure 6 materials-14-07772-f006:**
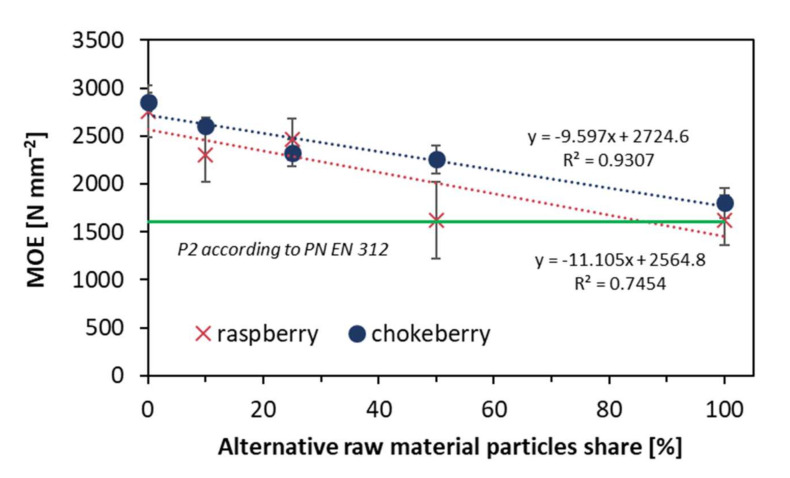
Modulus of elasticity of the tested composites.

**Figure 7 materials-14-07772-f007:**
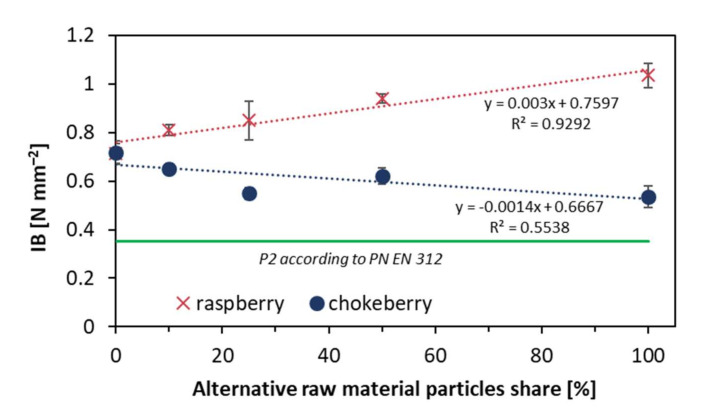
Internal bond of the tested composites.

**Figure 8 materials-14-07772-f008:**
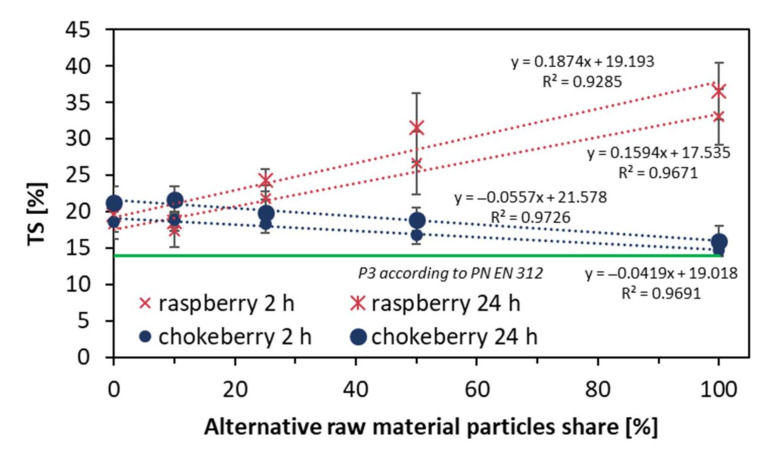
Thickness swelling of the tested composites.

**Figure 9 materials-14-07772-f009:**
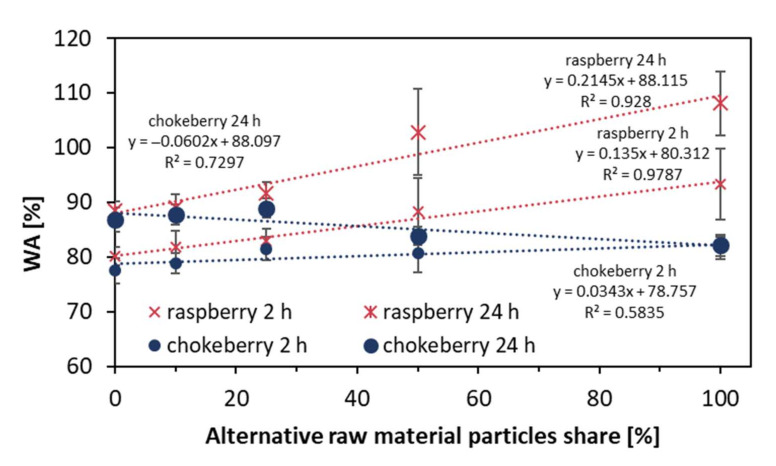
Water absorption of the tested composites.

**Figure 10 materials-14-07772-f010:**
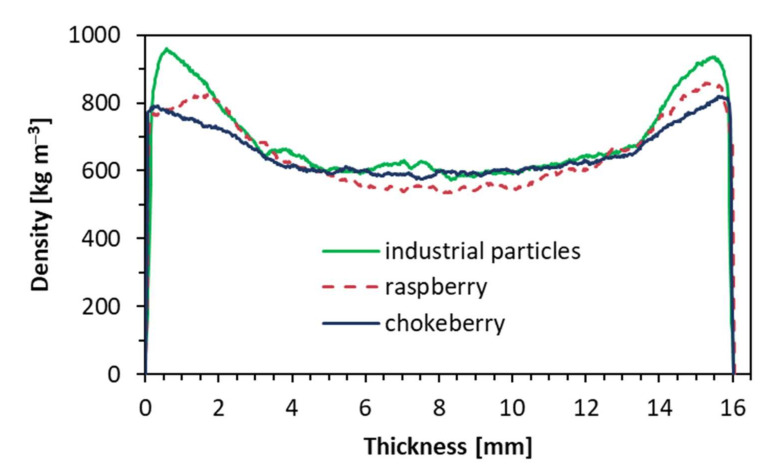
Density profiles of the tested composites.

**Figure 11 materials-14-07772-f011:**
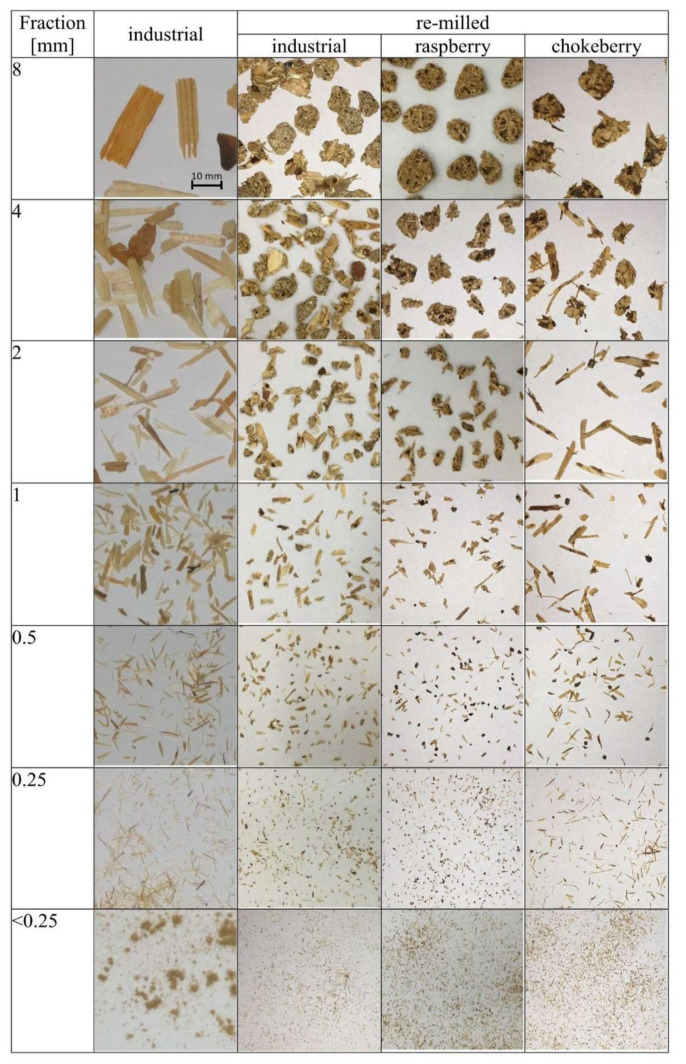
Pictures of the morphology of particles produced by re-milling the tested composites compared to industrial particles (each picture dimension is 50 mm × 50 mm).

**Figure 12 materials-14-07772-f012:**
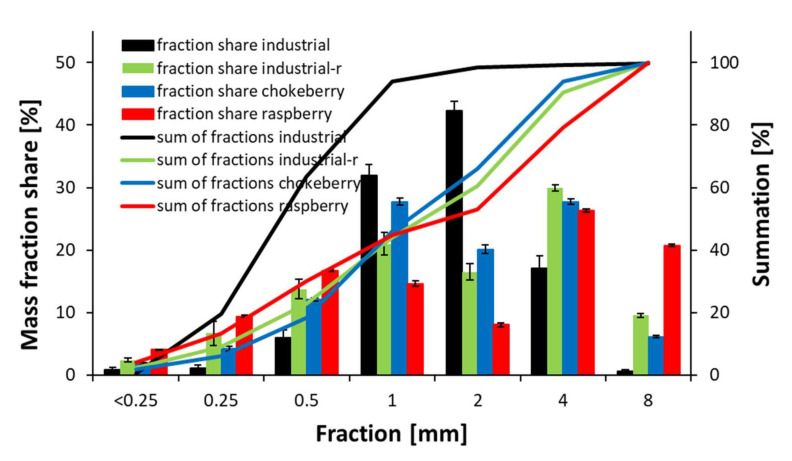
Mass fraction share of particles produced by re-milling of composites considering industrial particles.

## Data Availability

The data presented in this study are available on request from the corresponding author.

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
