# Peer review of "Upcycling and Recycling Potential of Selected Lignocellulosic Waste Biomass"

_materials, 2021, doi:10.3390/ma14247772_

Round 1

Reviewer 1 Report

The first sentence in abstract is The research aimed to confirm the ability to reduce carbon dioxide emissions by composite production using plantation waste on the example of lignocellulosic particles of black chokeberry (Aronia melanocarpa (Michx.) Elliott) and raspberry (Rubus idaeus L.)” I would like to know is there any calculation to confirm reduce of carbon dioxide emissions amount?

Results should be quantitatively reported in abstract.

Line 52 to 56, I do not think so it is necessary to explain about medical advantages of these lignocellulosic materials that much. Better to revise literature review related to wood-based panels.

The English structure of the paper should be more understandable to readers, also a few typing mistakes are existed in manuscript please carefully check it out.

I do not understand novelty degree of this study, if it is the first time the raspberry and chokeberry are used for particleboard production why in first step, hot-pressing variables did not optimize then go through mixing ratio?

Generally, vertical density profile should illustrate whole panel thickness, so it is better to change the graph.

However, the results presented here is very simple and fundamental. I would like to suggest the authors to add more experiments and in-depth discussion: compaction ratio of the produced particleboard and chemical components of raw material.

It is better figure 10 has scale bar on the picture. And for more understanding raspberry and chokeberry particles after chipping should add in manuscript.

Author Response

Dear Reviewer,
thank you for your comments. We tried to respond to them carefully (below).

Regarding the first sentence, the first sentence in the abstract “The research aimed to confirm the ability to reduce carbon dioxide emissions by composite production using plantation waste on the example of lignocellulosic particles of black chokeberry (Aronia melanocarpa (Michx.) Elliott) and raspberry (Rubus idaeus L.)”, this is a general description of an attempt to store as much as possible carbon dioxide in value-added products, as our panels can be, instead of burning the raw materials we used. In fact, these raw materials are currently burned. So, if we produce the panels out of raspberry or black chokeberry, the carbon dioxide fixed in these raw materials remains fixed (and not emitted to the atmosphere) for a longer time.

Results should be quantitatively reported in abstract.
We are afraid that due to the broad range of research and a limited number of words assigned to the abstract, it would be hard (or even impossible) to provide a quantitative report of the results in the abstract section. Thus, we decided to present results briefly in a qualitative way.

Line 52 to 56, I do not think so it is necessary to explain about medical advantages of these lignocellulosic materials that much. Better to revise literature review related to wood-based panels.
Apart from the literature review in the range of wood-based panels, we added two sentences regarding the medical advantages of the examined species to emphasize the potential of plantation of raspberry and black chokeberry in the context of availability of raw materials.

The English structure of the paper should be more understandable to readers, also a few typing mistakes are existed in manuscript please carefully check it out.
We have checked the manuscript carefully to fix any typographic mistakes.

I do not understand novelty degree of this study, if it is the first time the raspberry and chokeberry are used for particleboard production why in first step, hot-pressing variables did not optimize then go through mixing ratio?
So far there is no information in the literature about the utilization of these raw materials to produce particleboards. Regarding hot pressing, we applied the pressing parameters close to industrial conditions to have comparable results to those currently used, they happen to be the same recommendations made by resin/binder producers.

Generally, vertical density profile should illustrate whole panel thickness, so it is better to change the graph.
The graph has been changed.

However, the results presented here is very simple and fundamental. I would like to suggest the authors to add more experiments and in-depth discussion: compaction ratio of the produced particleboard and chemical components of raw material.
Since we just started the examination of mentioned raw materials in the context of the production of lignocellulosic composites (no such research in literature), we intend to go deeper into the issues you mentioned in further research. Anyway, thank you for this valuable remark!

It is better figure 10 has scale bar on the picture. And for more understanding raspberry and chokeberry particles after chipping should add in manuscript.
The scale bar has been added to the picture.

In the attachment please find the corrected manuscript.

Thank you!

Reviewer 2 Report

In my opinion manuscript materials-1500330 can be improved in view of publication as follows:

1.The title is too general, bio waste should be named.

2.Novelty and work hypothesis should be better explained in the introduction and in the abstract. The introduction is too broad, too general. presentation of literature data on similar particle boards should be emphasized.

3.Figures should be given close to their citation in the text. Thus, Fig 1 should be in sections 2.1., etc

4.Figure 10 at line 207 should have no. 6. Figure numbering and order should be checked.

5.English revision and typos correction is recommended: line 99, was should be were, line 184 The only should be the only.

6.Results and discussion text body should be organized in subsections for  a better presentation. More comparisons with similar data  should be made in this section. Was the effect of particle boards natural aging considered ?

Author Response

1. The title is too general, bio waste should be named.
The title has been changed.

2. Novelty and work hypothesis should be better explained in the introduction and in the abstract. The introduction is too broad, too general. presentation of literature data on similar particle boards should be emphasized.
The novelty character range of research has been emphasized in the abstract. The hypothesis has been added. We referred the introduction literature review to particleboards made of alternative raw materials, as it was possible; however, there is no similar research about the application of raspberry and/or chokeberry in particleboards production.

3. Figures should be given close to their citation in the text. Thus, Fig 1 should be in sections 2.1., etc
Since fig. 1 shows the result of microscope analysis of the structure of investigated raw materials, we placed this figure in the Results and discussion section (and not in the methodology description section).

4. Figure 10 at line 207 should have no. 6. Figure numbering and order should be checked.
The fig. 10 cited in line 207 refers to the pictures of the particles since in mentioned sentence the description of the influence of the shape of the particles on MOR and MOE is provided. However, we accept the editorial re-distribution of the pictures among the manuscript wherever needed.

5. English revision and typos correction is recommended: line 99, was should be were, line 184 The only should be the only.
We corrected the mentioned errors and re-checked the entire manuscript. Thank you!

6. Results and discussion text body should be organized in subsections for  a better presentation. More comparisons with similar data  should be made in this section. Was the effect of particle boards natural aging considered ?
The sub-sections have been defined in the Results and Discussion section. We did our best to refer our results to similar data in the literature, but the literature about the utilization of such unconventional raw materials is very limited. No, the effect of natural aging on tested particleboards has not been investigated, but we plan to do it in further research.

In the attachment please find the corrected manuscript.

Thank you!

Reviewer 3 Report

The study "Upcycling and Recycling Potential of Selected Lignocellulosic Biopolymer Raw Materials" is interesting for the audience, has a solid structure and the methods used are efficiently applied. After revising the points mentioned below, the paper can be recommended for publication. 

Out of total of 37 sources, there are only 7 current sources from 2018 to 2021. Therefore, the Introduction part needs to be expanded and current studies that deal with the topic of the study and report on other lignocellulosic materials should be included. 

For example, the following studies would fit well:  
https://doi.org/10.1016/j.indcrop.2017.12.074
https://doi.org/10.1016/j.matpr.2020.06.009

In the materials and methods part, exact details of materials, machines and equipment (manufacturer, model, country of manufacturer etc) used in this study should be added. 
Line 71: How were raw materials dried and under what conditions?
Line 73: The wooden branches of chokeberry and raspberries were shredded, how?
Line 89: At this point should be represented as 10%, 25%, 50%, and 100%.
Line 90: From which manufacturer and country do urea-formaldehyde resin come?
Line 92: Which oven exactly was used? The exact information (manufacturer, model, country of manufacturer, etc.) is missing here. 
Line 93: With which equipment were panels pressed?
Line 97: Which device was used to cut particleboards?
Line 109: Details (manufacturer, model, country of manufacture, etc.) of the laboratory knife mill are missing.
Line 111: Information about manufacturer, model, country of manufacture, etc. for IMAL vibrating laboratory sorter is missing. 
Figure 3: Please check the SD, if they are correct, because Chokeberry (face-164) is going in, which can't be possible. 
Line 184: Please correct the sentence. 
If possible, please insert the CLSM or light microscope images with a larger magnification to show the surface morphology of chokeberry and raspberry. 

Figure 10 is actually a table. This needs to be formatted better. 

A table with a sample overview should be shown for better clarity.

The process of sample preparation/production should be shown graphically to give a clearer overview. Currently, it is difficult to understand how these were prepared. 

Author Response

The study "Upcycling and Recycling Potential of Selected Lignocellulosic Biopolymer Raw Materials" is interesting for the audience, has a solid structure and the methods used are efficiently applied. After revising the points mentioned below, the paper can be recommended for publication.
Thank you!

Out of total of 37 sources, there are only 7 current sources from 2018 to 2021. Therefore, the Introduction part needs to be expanded and current studies that deal with the topic of the study and report on other lignocellulosic materials should be included.
For example, the following studies would fit well:
https://doi.org/10.1016/j.indcrop.2017.12.074
https://doi.org/10.1016/j.matpr.2020.06.009
The suggested papers have been cited. Thank you for your help with state of art update.

In the materials and methods part, exact details of materials, machines and equipment (manufacturer, model, country of manufacturer etc) used in this study should be added.
Line 71: How were raw materials dried and under what conditions?
Line 73: The wooden branches of chokeberry and raspberries were shredded, how?
Line 89: At this point should be represented as 10%, 25%, 50%, and 100%.
Line 90: From which manufacturer and country do urea-formaldehyde resin come?
Line 92: Which oven exactly was used? The exact information (manufacturer, model, country of manufacturer, etc.) is missing here.
Line 93: With which equipment were panels pressed?
Line 97: Which device was used to cut particleboards?
Line 109: Details (manufacturer, model, country of manufacture, etc.) of the laboratory knife mill are missing.
Line 111: Information about manufacturer, model, country of manufacture, etc. for IMAL vibrating laboratory sorter is missing.
Figure 3: Please check the SD, if they are correct, because Chokeberry (face-164) is going in, which can't be possible.
Line 184: Please correct the sentence.
If possible, please insert the CLSM or light microscope images with a larger magnification to show the surface morphology of chokeberry and raspberry.
The above-mentioned remarks have been included in the manuscript. Regarding microscope images – at the moment it is hard to get more pictures of surface morphology.

Figure 10 is actually a table. This needs to be formatted better.
The individual pictures in figure 10 have been organized in a table to simplify the description and make the overview more transparent. However, we prefer to consider it as a figure instead of the table.

A table with a sample overview should be shown for better clarity.
We prefer to keep the text description of the sample overview instead of a table since the suggested table content will be the redundant repeating of the information provided in the text.

The process of sample preparation/production should be shown graphically to give a clearer overview. Currently, it is difficult to understand how these were prepared.
The suggested graphic has been added.

In the attachment please find the corrected manuscript.

Thank you!

Round 2

Reviewer 3 Report

The authors have revised the manuscript well. Only table 10 should be reformatted better because the numbers in the first column stick to the margin.